# Research Status and Progress of Welding Technologies for Molybdenum and Molybdenum Alloys

**Qi Zhu [1], Miaoxia Xie [2,\*], Xiangtao Shang [2], Geng An [1,3], Jun Sun [3], Na Wang [1], Sha Xi [1], Chunyang Bu [1] and Juping Zhang [1]**

1   Technical Center, Jinduicheng Molybdenum Co., Ltd., Xi'an 710077, China; zhuyaqian2009@163.com (Q.Z.); gmail@163.com (G.A.); biyewangna@163.com (N.W.); xs19861105@126.com (S.X.); bp208@163.com (C.B.); hjsjdc@yeah.net (J.Z.)
2   School of Mechanical and Electrical Engineering, Xi'an University of Architecture and Technology, Xi'an 710055, China; shang_x_t@163.com
3   State key laboratory for mechanical behavior of materials, Xi'an Jiaotong University, Xi'an 710049, China; junsun@mail.xjtu.edu.cn
\*   Correspondence: Xiemiaoxia@xauat.edu.cn; Tel.: +86-181-4902-5125

**Abstract:** Owing to its potential application prospect in novel accident tolerant fuel, molybdenum alloys and their welding technologies have gained great importance in recent years. The challenges of welding molybdenum alloys come from two aspects: one is related to its powder metallurgy manufacturing process, and the other is its inherent characteristics of refractory metal. The welding of powder metallurgy materials has been associated with issues such as porosity, contamination, and inclusions, at levels which tend to degrade the service performances of a welded joint. Refractory metals usually present poor weldability due to embrittlement of the fusion zone as a result of impurities segregation and the grain coarsening in the heat-affected zone. A critical review of the current state of the art of welding Mo alloys components is presented. The advantages and disadvantages of the various methods, i.e., electron-beam welding (EBW), tungsten-arc inert gas (TIG) welding, laser welding (LW), electric resistance welding (ERW), and brazing and friction welding (FW) in joining Mo and Mo alloys, are discussed with a view to imagine future directions. This review suggests that more attention should be paid to high energy density laser welding and the mechanism and technology of welding Mo alloys under hyperbaric environment.

**Keywords:** molybdenum alloy; welding; status; progress

## 1. Introduction

Molybdenum (Mo) and Mo alloys show characteristics such as high melting point, good high-temperature strength, high wear resistance, high thermal conductivity and low resistivity, low coefficient of linear expansion, high elastic modulus, and good corrosion resistance [1]. Based on this, they have irreplaceable functions and application demands in the fields like the defense industry, aerospace, electronic information, energy, chemical defense, metallurgy, and nuclear industry. However, Mo and Mo alloys are hard and brittle materials in nature, so their weldabilities are generally poor [2]. There are two main sources of molybdenum brittleness: one is the intrinsic brittleness of molybdenum, and the other is the enrichment of interstitial impurities in the grain boundary. Oxygen is the most important impurity element in the grain boundary which affects the embrittlement of molybdenum. The solubility of oxygen in molybdenum at room temperature is less than 0.1 ppm, which forms relatively volatile molybdenum oxide at the grain boundary, which greatly reduces the bond strength of grain boundary. After melting and welding of high-performance molybdenum alloy, the weld forms

an as-cast structure with coarse grains, the heat-affected zone forms a coarse recrystallization structure, the impurity elements are enriched in the grain boundary, and the strength and toughness of the weld and heat-affected zone are greatly reduced [3–5]. To extend application field of Mo and Mo alloys, researchers worldwide have conducted a lot of studies on their welding and relevant kinds of literature have been reported since the 1970s [1].

In recent years, the global nuclear industry and scientific community have been aware that a new fuel system, that is, accident tolerant fuel (ATF) needs to be developed [6,7]. Such a fuel system needs to be able to withstand severe accident conditions and slow down the rate of deterioration over a long period of time, to provide more valuable time for people to take emergency measures and greatly reduce the risks of leakage of radioactive materials. Therefore, Mo alloy is listed as one of the main candidate materials for ATF cladding by the global nuclear industry [8]. In this context, the welding technologies for Mo and Mo alloys have attracted wide attention of researchers in China and a lot of new progress has been achieved in recent years.

## 2. Analysis on Weldability of Mo and Mo Alloy

### 2.1. Room-Temperature Brittleness

Ductility of most of Mo alloys varies with temperature to make the materials change from ductile fracture to brittle fracture in a very small temperature range. The ductile-brittle transition temperature of pure Mo ranges from approximately 140 °C to 150 °C, resulting in difficulties in intensive processing, low product performance, and limited application fields [1]. Such brittleness is known as intrinsic brittleness of Mo, which is mainly determined by an electron distribution characteristic that the outermost and sub-outermost electrons of its atoms are half full.

Mo and Mo alloy have a high melting point, good thermal conductivity, high recrystallization temperature, no allotropy transformations in solid state, and low density of bcc crystal structure. Due to these characteristics, weld seam and heat-affected zone (HAZ) is large and grains are seriously coarsened after welding (Figure 1), so that interstitial impurities, such as C, N, and O are fully diffused and enriched on grain boundaries, resulting in greatly weakened bonding strength of grain boundaries, like Figure 2 that the fracture of laser welded Mo alloy contains a lot of $MoO_2$. Under the joint effects of intrinsic brittleness of the materials and segregation of impurities at grain boundaries, sensitivity of welding cracks is high and strength, plasticity, and ductility of Mo and Mo alloy joints are poor [9,10]. Therefore, molybdenum and molybdenum alloy parts or structures are usually manufactured by powder metallurgy rather than welding.

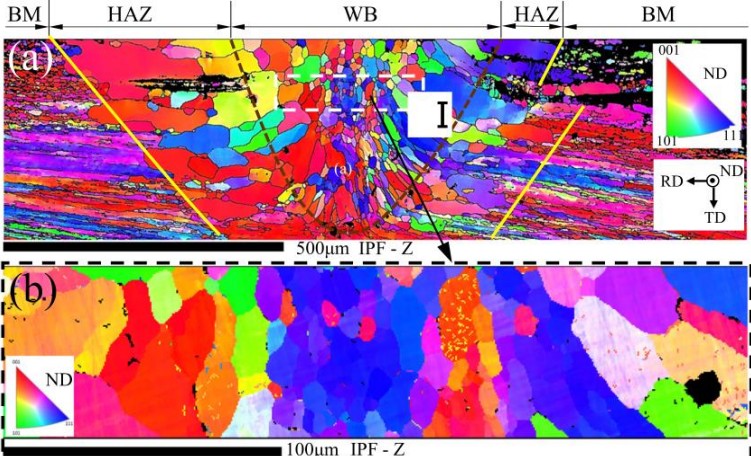

**Figure 1.** (**a**) Electron backscatter diffraction (EBSD) image of cross section of Mo joint achieved by laser welding, and (**b**) enlarged view of area (**a**) [10].

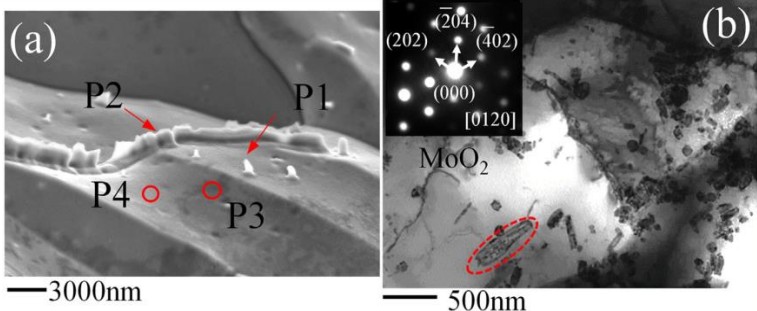

**Figure 2.** (**a**) SEM image of the fracture of Mo joint with plenty of MoO$_2$ at grain boundary, and (**b**) TEM bright field image of the weld bead of the Mo joint [10].

### 2.2. Pore Defects

Furthermore, owing to the powder metallurgy process that can yield fine grain structure without preferred orientation, the refractory metal work blanks are usually prepared by utilizing powder metallurgy method. This leads to that the material contains micropores and impurity elements, and its compactness is not comparable to that of the materials produced by smelting metallurgy. Therefore, the problem of high rate of pore defects (Figure 3) generally appears in fusion-welded Mo and Mo alloys. Particularly, the high-pressure residual gases in the micropores are the most harmful. During the welding process, these high-pressure gases can expand rapidly in the molten pool after being released into the high-temperature molten pool, which seriously deteriorates the quality of welded joints of Mo and Mo alloys [11–13].

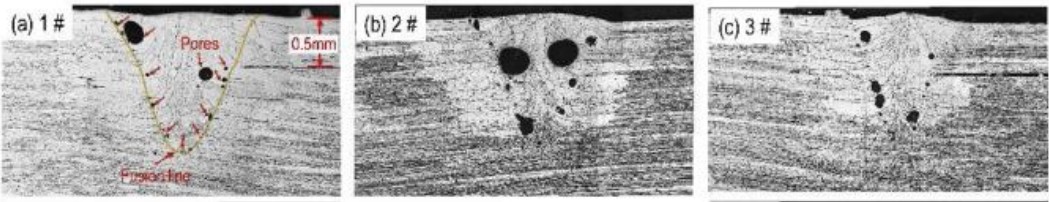

**Figure 3.** Typical porosity morphology in Mo joints produced by laser welding: (**a**) orthogonal experiment 1# parameter sample, (**b**) orthogonal experiment 2# parameter sample, and (**c**) orthogonal experiment 3# parameter sample. [11].

### 3. Research Progress in Welding of Mo and Mo Alloys

At present, the welding methods for Mo and Mo alloys mainly include tungsten-arc inert gas (TIG) welding, electron-beam welding (EBW), laser welding, electric resistance welding (ERW), brazing, and friction welding.

### 3.1. EBW

Pan et al. [14] studied EBW for pure Mo with a thickness of 1.5 mm obtained by powder metallurgy. The results show that the faster the welding speed is, the small the grain size and the less the interstitial impurities. By increasing welding speed and reducing welding heat input, the ductility of welded joint of Mo can be significantly improved. Vacuum degree significantly affects ductile-brittle transition temperature, whereas the decomposition of oxides on the surface of the workpiece during welding has great influence on the vacuum degree. When a vacuum degree increases from 10−4 mm Hg to 10−5 mm Hg, the upper limit of ductile-brittle transition temperature of the welded joint decreases from ~150 °C to ~100 °C. Pan et al.'s analysis of the change in joint performance through the impurity composition and impurity content of the gas released after the melting of the material left a deep impression on the authors. Yang et al. [15] welded pure Mo plates with thickness of 16 mm by using

EBW. The results demonstrate that the highest strength of the welded joint subjected to heat treatment at 1100 °C was found in the weld seam. Tensile fractures of the welded joint are shown in the weld seam and present morphology of cleavage fracture. In addition, Zheng et al. [16] welded pure Mo materials with thickness of 16 mm through vacuum EBW. As can be seen from the above research results, the results illustrate that grains in weld seam of Mo by EBW grow rapidly.

Morito et al. (1989) [17] found that a joint of molybdenum-titanium-zirconium (TZM) alloys welded using EBW at room temperature always shows brittle fracture. However, at temperature higher than 300 °C, the joint always presents ductile fracture, and there is obvious necking phenomenon before fracturing. In addition, the research demonstrates that carbonization and heat treatment after welding can effectively raise the strength of the joint of Mo alloys, which is primarily attributed to an increase of the grain boundary cohesion due to the effective carbon segregation and precipitation. [18]. Morito et al. (1998) [19] found that the strength and ductility of a welded joint of Mo alloys obtained through EBW can rise after increasing rhenium (Re) content. The reason is that two-phase microstructure is formed in the weld zone with enhanced Re content. Based on the thermal simulation test, Morito et al. (1997) [20] compared ductilities of HAZs when welding Mo alloys (Mo > 99.9 wt%) under two conditions of thermal treatment, i.e., furnace cooling and quick cooling through quenching. It is found that quick cooling after welding can significantly reduce the ductility of HAZ of the welded joint of Mo alloys, mainly because grain-boundary segregation in HAZ is more significant under quick cooling through quenching. Stütz et al. (2016) [21] systematically studied the influences of parameters of EBW process on sizes of fusion zone (FZ) and HAZ, size of grains in FZ and HAZ, sensitivity to pores and cracks in butt welded joint of TZM alloy with thickness of 2 mm. The results show that pore defects are serious when welding heat input is large. Small heat input can not only inhibit pores, but also obviously reduce grain size in FZ. The strength of the joint welded by EBW can reach 50–77%: that of the base metal (BM). Therefore, Stütz et al. pointed out that it was necessary to study EBW in terms of filling materials and alloying metals in the weld seam, which has important guiding significance for improving mechanical properties of molybdenum alloy joints. Recently, Chen et al. (2018) [22] conducted electron beam welding of molybdenum and found that the even tensile strength of the joints was 280 MPa, and the fracture position was located in the weld, which was a brittle fracture, the fracture location is shown in Figure 4. It was determined as quasi-cleavage fracture. Pore and crack defects were observed in the weld zone. The pores were formed by the oxygen that was not escaping the molten pool. Cracks were confirmed as solidification cracks and low plastic embrittlement cracks.

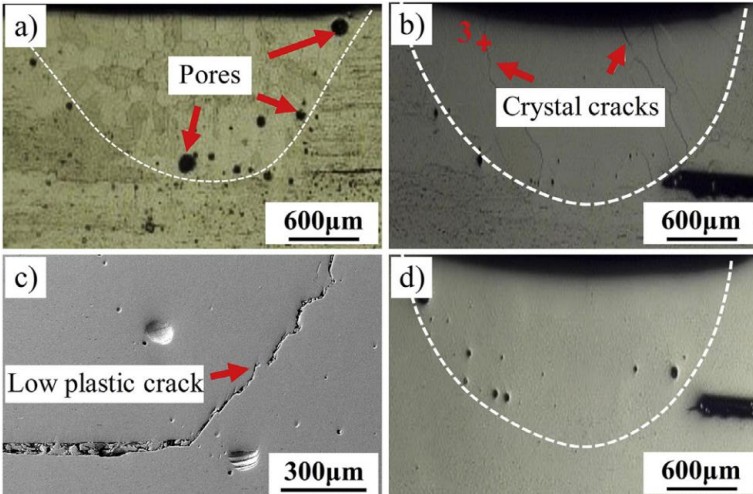

**Figure 4.** Cross-sectional morphology of the welded joints produced by electron-beam welding (EBW) welding (**a**) pore defects diagram with improved parameters, (**b**) crystal cracks, (**c**) low plastic embrittlement crack, and (**d**) forming after optimized process. [22].

In addition, Chen et al. (2019) [23] found that during welding with a 0.6 mm beam deflection to Kovar, the weld zone exhibits equiaxed crystals. Compared with the columnar crystal during welding without beam deflection, the microstructure of the weld zone transforms into an equiaxed crystal when welding with a 0.6 mm beam offset to Kovar. The morphology of the reaction layer alters due to the beam deflection, which escalates its toughness. No cracks are observed in the heat-affected zone on the molybdenum side. The tensile strength of the joint increases resulting from the beam deflection to Kovar, which exceeds 260 MPa when the beam deflection is 0.6 mm.

### 3.2. TIG Welding

Wang et al. [24] studied the TIG welding of TZM alloy. The results demonstrate that well-formed weld seam can be obtained under proper welding current (as is shown in Figure 5), welding speed, and argon gas flow, better welding process parameters: welding speed 4 mm/s, argon flow rate 10 L/min, welding current should be controlled at ~210 A. Jiang et al. [25] researched TIG welding of Mo-Cu composite materials and stainless steel filled with Cr-Ni wires.

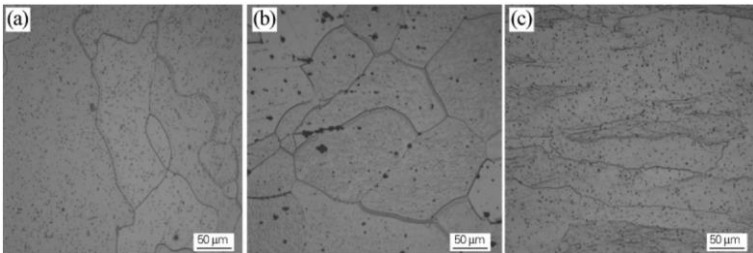

**Figure 5.** Microstructure of weld joint (**a**) welding seam zone, (**b**) heat-affected zone, and (**c**) TZM martix. [24].

Wang [26] found that there are fewer pores in weld seams obtained by EBW or TIG welding of smelting Mo alloy, whereas there are more pores in weld seams of pure Mo or Mo alloys obtained by powder metallurgy. The addition of C can improve plasticity of powder-metallurgy-processed weld seam of pure Mo or Mo alloys by reducing content of molybdenum oxide at grain boundary, and significantly reduce pores in the weld seam. In addition, adding Ti and hafnium (Hf) into the weld seam can decrease centerline cracks and pores in weld seam of powder-metallurgy-processed pure Mo, so that tensile fracture transfers from weld seam to HAZ.

Matsuda et al. [27] studied EBW and TIG welding of TZM alloy with thickness of a 1.5 mm prepared through powder metallurgy. The research shows that large welding heat input can significantly decrease ductility of a welded TZM alloy joint and the ductile-brittle transition temperature of the welded joint obtained through TIG welding is ~120 °C higher than that of EBW. Furthermore, they also found that pore defects only appear around the arc starting and arc extinguishing positions in weld seam during TIG welding, whereas pore defects greatly increase in weld seam during EBW welding in the vacuum environment. Kolarikova et al. (2012) [28] investigated EBW and TIG welding of pure Mo sheets. Widths of FZs in joints obtained through EBW and gas tungsten arc welding (GTAW) separately are 0.8 mm and 1.7 mm, whereas HAZs are significantly different in width (1.4 mm and 35 mm). Grain sizes in FZ and HAZ in the EBW joint are obviously smaller than those in the GTAW joint, indicating that EBW with high energy density is more suitable for welding Mo than GTAW

### 3.3. Laser Welding

Liu et al. (2016) [29] studied continuous-wave Nd:yttrium-aluminum-garnet (YAG) laser welding of an overlap joint of Mo-Re alloy (50Mo:50Re) with the thickness of 0.13 mm prepared by powder metallurgy. After welding, cracks appear at the bonding interface of FZ, and many large pores are observed at the bonding interface of FZ, with the diameter being ~15–20% of the thickness of BM plates. Microscopic analysis results of fracture present that intergranular fracture occurs and there are

a large number of dark compounds in the grains and on the grain boundary, as is shown in Figure 6, and the composition analysis showed that the content of C and O in these compounds were 30% and 15%, respectively. It is believed that coarse microstructures and harmful impurity elements cause the hardening of bonding interface and intergranular fracturing of the joint. The study of Lin (2013) [3] demonstrates that welding conductive elements of needle-shaped pure Mo with a diameter of 0.5 mm using pulse Nd:YAG laser welding instead of ERW can significantly raise the strength of the joint.

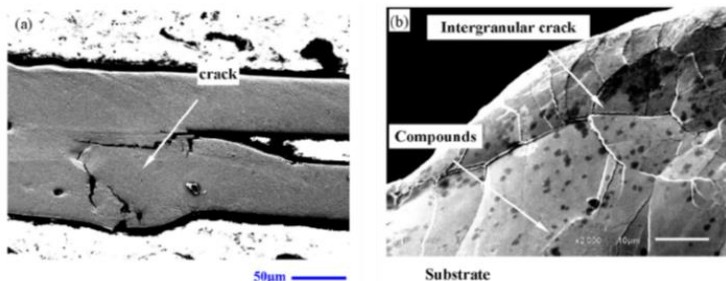

**Figure 6.** SEM observation of 50Mo-50Re overlap joint achieved by laser welding: (**a**) cross section and and (**b**) fracture surface [29].

Kramer et al. (2013) [4] studied EBW and pulse Nd:YAG laser welding of Mo-44.5% Re alloy sheets with a thickness of 0.5 mm. The research shows that a Mo-44.5% Re alloy joint welded by EBW is well-formed, without defects including pores and cracks. Cracks are found in FZ in the laser-welded Mo-44.5% Re alloy joint and the micromorphology of brittle fracture is shown in the fracture of the laser-welded joint after testing mechanical performances. Chatterjee et al. (2016) [5] researched EBW and Nd:YAG laser-TIG hybrid welding for butt welded joint of wrought TAM alloy (Ti 0.50 wt%, Zr 0.08 wt% and C 0.04 wt%) with thickness of 1.2 mm. Grain sizes in FZ and HAZ in the EBW joint are obviously small, which are ~55% and ~65% of those in the joint obtained by Nd:YAG laser-TIG hybrid welding. The weld width of EBW and hybrid welding method is ~1.4 mm and 2.6 mm respectively, but in both cases, the width of HAZ is ~1.5 times of the weld width. Results of tensile test demonstrate that strengths of joints prepared by Nd:YAG laser-TIG hybrid welding and EBW are about 41% and 47% that of BM. The fracture morphology is shown in Figure 7, the two joints hardly show any tensile plasticity in the tensile test, and the shrinkage and elongation of cross sections are almost zero, while the elongation of BM is up to 8.4%. Figure 8 shows the presence of large grains and nearly uniform distribution of a second phase within the gain, typical volume fraction estimated from these micrographs shows that the oxide phase is nearly 10 pct of the volume fraction.

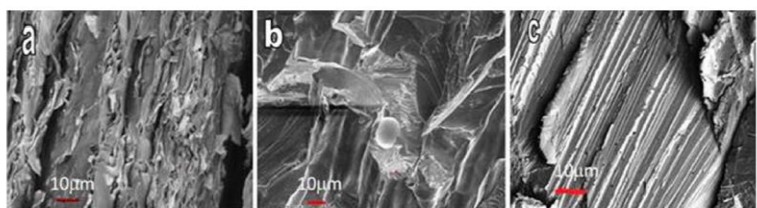

**Figure 7.** Fractographs of (**a**) the parent metal, (**b**) EB weld joint, and (**c**) LGTHW Joint of tensile samples. Change in the morphology and the presence of sharp faceting in samples containing weld joints could be noticed [5].

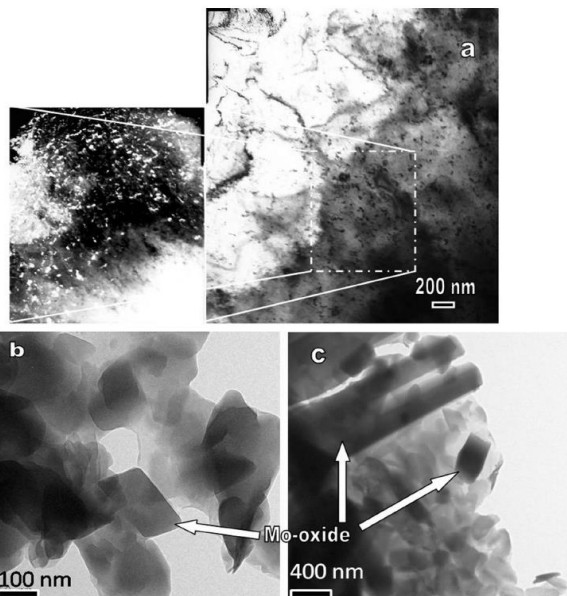

**Figure 8.** TEM micrographs showing (**a**) the presence of the precipitates within the matrix. Inset shows the magnified view; (**b**) Magnified view showing the presence of Mo-oxide phase in the weld region. (**c**) The presence of needle-shaped long oxides may be noticed [5].

An et al. (2018) [30] carried out laser lap welding of fuel cladding and end plug made of molybdenum (Mo) alloy. Under the optimum processing conditions, the tensile strength of the welded joint reached 617 MPa, taking up 82.3% that of the base metal. Recently, Zhang et al. (2019) [9] successfully enhanced the mechanical performance of fusion zone in laser beam welding joint of molybdenum alloy by solid carburizing. As is shown in Figure 9 the tensile strength of carburized weld joints rose by 426% compared with that of uncarburized weld joints. The TEM images of molybdenum oxide particles in the FZ of LW joint show that the particles were shown as lenticular or elongated blocks under TEM observation (Figure 10a) and judged from the diffraction pattern as $MoO_2$. And the TEM images of molybdenum carbide particles in the FZ of SC-150 joint show that the particles were shown as circular pattern under TEM observation (Figure 10b) and judged from the diffraction pattern as $Mo_2C$.

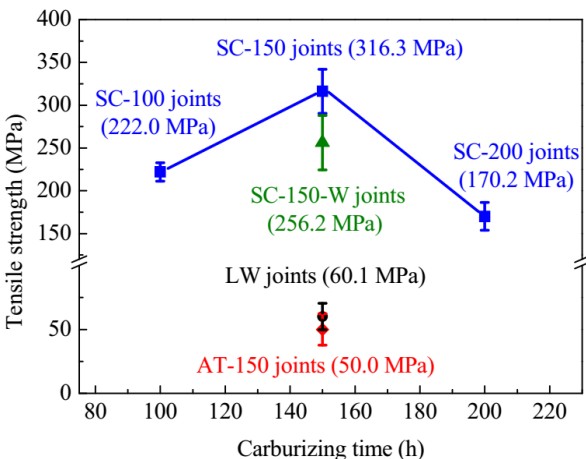

**Figure 9.** The effect of C addition on tensile strength of the Mo alloy joints [9].

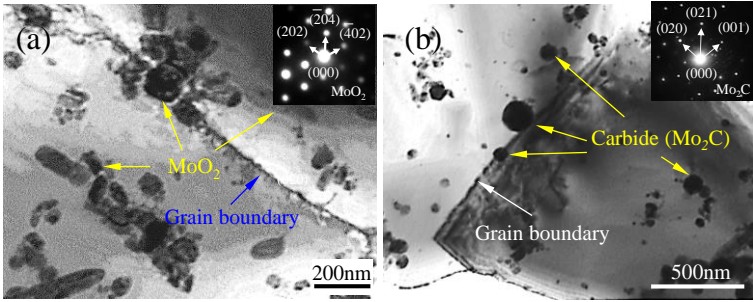

**Figure 10.** TEM images of (**a**) molybdenum oxide particles in the FZ of LW joint and (**b**) carbide particles in the FZ of SC-150 joint [9].

Xie et al. (2019) [12] found that low heat input (i.e., high welding speed) resulted in significantly refined grains in the fusion zone (FZ) of fiber laser welded nano-sized rare earth oxide particles and superfine crystal microstructure (NS) Mo joints, the cross-sectional microstructures of the NS Mo alloy laser welding joints is shown in Figure 11. When welding heat input decreased from 3600 J/cm (i.e., 1.2 kW, 20 cm/min) to 250 J/cm (i.e., 2.5 kW, 600 cm/min), the tensile strength of welded joints increased from ~250 MPa to ~570 MPa. They also found that laser welding of NS-Mo under low heat input significantly reduced the porosity defects in the fusion zone (2019) [13].

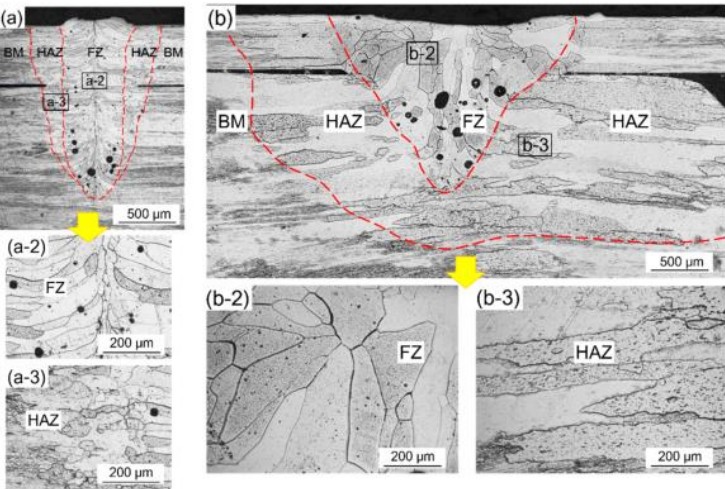

**Figure 11.** The cross-sectional microstructures of the NS Mo alloy laser welding joints. Heat input: (**a**) 250 J/cm and (**b**) 3600 J/cm [12].

In the study of Zhang et al. (2019) [10], titanium was selected as an alloying element to reduce brittleness of laser weld beads in molybdenum "cladding-end plug" socket joints. Brazing was also performed to enhance joint strength. The combined structure of laser welding and brazing is shown in Figure 12, joints with the same strength as base material and hydraulic bursting pressure of 60 MPa were produced using a combination of the two methods.

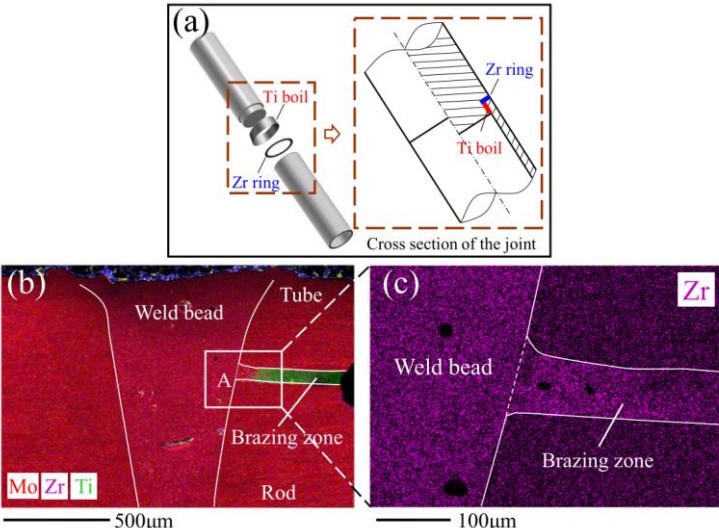

**Figure 12.** Tracing of the metal elements in the weld bead of the Mo-0.03Ti-B joint: (**a**) schematic diagram of the location where Zr was added. (**b**) Analysis of Mo, Zr, and Ti on the cross section of the joint by EDS surface scanning. (**c**) Analysis of Zr in zone A in panel (**b**) by EDS surface scanning. [10].

Liu et al. (2019) [11] found that welding cycles had a significant influence on the porosity ratio of fusion zone (FZ), whereas the amplitude and frequency of laser power waveform slightly influenced the porosity. Moreover, Zr added in a molten pool can be preferentially reacted with O to generate $ZrO_2$, which can inhibit the precipitation of volatile $MoO_2$ to thus suppress the generation of metallurgy-induced pores, reconstructed 3D transparent distribution of pores in joints is shown in Figure 13. Zhang et al. (2019) [7] conducted laser seal welding of end plug to thin-walled nanostructured high-strength molybdenum alloy cladding with a zirconium interlayer and tensile strength of the achieved welded joints matched that of the base metal. Note that by taking advantage of the metallurgical characteristics of molybdenum and its high melting point and high thermal conductivity, Zhang et al. [9–11] put forward a systematic strategy that can effectively solve the problems of porosity and embrittlement in welding fuel cladding made of molybdenum alloys. More importantly, the results of tensile test and hydraulic bursting show that the molybdenum alloy cladding prepared by this method have excellent performance, which completely eliminates the doubts about the welding quality of molybdenum alloy fuel cladding, and is of great significance to the promotion and application of molybdenum alloy accident-tolerant fuel cladding, the results of tensile test and hydrostatic test are shown in Figure 14.

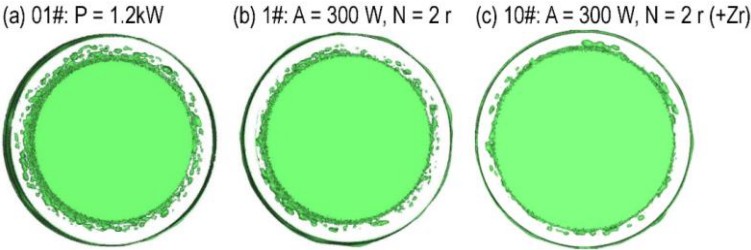

**Figure 13.** Reconstructed 3D transparent distribution of pores in the three joints achieved under (**a**) P = 1.2 kW; (**b**) average power = 1.2 kW, Amplitude = 300 W, frequency = 50 HZ, N = 2 cycles; and (**c**) average power = 1.2 kW, amplitude = 300 W, frequency = 150 HZ, N = 2 cycles, adding Zr, respectively. [11].

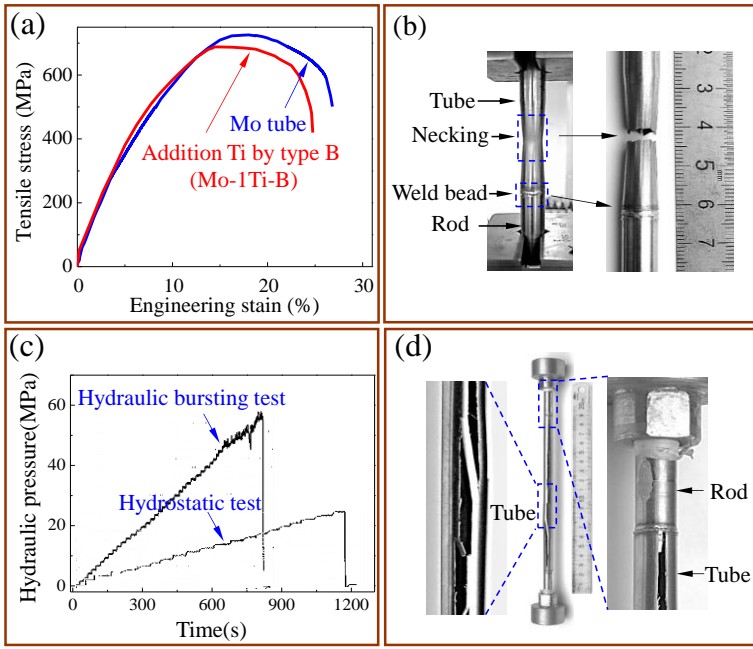

**Figure 14.** (**a**) Strength-displacement curves of Mo-0.03Ti-B and pure Mo tube; (**b**) images of the joint after fracture testing; (**c**) hydrostatic test curve and hydraulic bursting test curve for the molybdenum joint of Mo-0.03Ti-B; and (**d**) images of joint after hydraulic bursting testing [10].

In the study of Wang et al. (2019) [31], laser beam offset welding was used to join pure Mo and 304L. The results demonstrated that tensile strength of the joints could be increased to ~280 MPa by presetting a nickel (Ni) foil at Mo/304 L interface and shifting laser beam to 304L, whereas the tensile strength of the sample without Ni foil is only 112 MPa, and the results of tensile test is shown in Figure 15. In the study of Lu et al. (2018) [32,33], by adding zirconium (Zr) to the molten pool, ultimate tensile strength (UTS) of the dissimilar joint of titanium and molybdenum was increased from about 350 MPa to ~470 MPa, which reached more than 90% of that of the Ti base metal (BM). Zhou et al. (2018) [34] observed cracks in dissimilar laser welding of tantalum to molybdenum and pointed out that solidification cracking tendency of Mo was the main reason for crack initiation in the Ta/Mo joint. Ning et al. (2019) [35] studied the potential of laser welding of 0.5 mm-thick Titanium-zirconium-molybdenum (TZM) alloy in a lap welding configuration. They found that introducing an interface gap of 0.09 mm had the most positive effect in reducing the porosity compared to using helium gas, different shielding gas flow rates, adding alloy elements, and different heat input rates. Liu et al. (2019) [36] compared the micro-structures, properties, and residual stresses of the welded girth joints achieved at different preheating temperatures and found that the tensile strength reached a maximum at the preheating temperature of 673 K, which was approximately 50% that of the base metal. Gao et al. (2020) [37] also studied the effect of laser offset on microstructure and mechanical properties of laser welding of pure molybdenum to stainless steel. As the laser beam shifts from the Mo side to the stainless steel side, the formation of welding defects and Fe-Mo intermetallic compounds (IMCs) are effectively restricted because of the decrease amount of molten Mo. Consequently, the tensile strength of joints increased first and then decreased in the laser offset range of 0.2–0.5 mm. The highest tensile strength of the joints is 290 MPa at the laser offset of 0.3 mm.

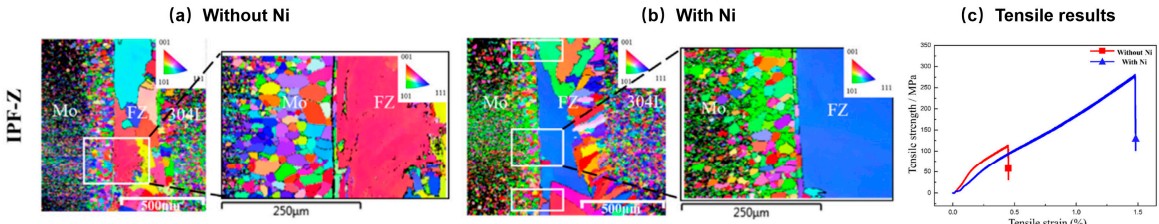

**Figure 15.** (**a**) Fracture cross section of the joint without Ni, (**b**) fracture cross section of the joint with Ni, and (**c**) tensile results of the two joints [31].

### 3.4. ERW

Xu et al. (2007) [38] investigated the optimization of the resistance spot welding process of an overlap joint of 50Mo-50Re (wt%) alloy with thickness of 0.127 mm. The study found that the longer the time of application of upsetting force after power off is, the higher the strength of the joint and the better the ductility. When time of applying upsetting force after power off increases from 50 ms to 999 ms, bearing capacity of the joint rises from 100 N to 113 N and fracture mode changes from brittle intergranular fracture to dimple fracture. This is because after power off, the increase of time of applying upsetting force can accelerate the cooling rate of weld seam, thus inhibiting segregation of Mo at grain boundaries. It is also found from the study that the defects of large-size pores appear in FZ of the joint under various welding conditions, because there are micropores in powder metallurgy materials and is shown in Figure 16.

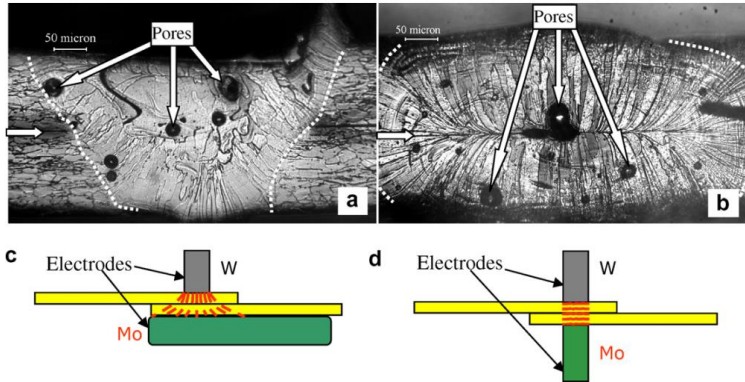

**Figure 16.** (**a**,**b**) The shape and microstructure of the nuggets welded using the electrodes shown in (**c**,**d**), respectively. The horizontal arrows show the interfaces between two workpieces in panels (**a**) and (**b**) [38].

Elizabeth E. Ferrenz et al. [39] controlled welding quality by using double pulse current waveforms in resistance spot welding of Mo and T-Re alloy wires. The first pulse current is small and mainly used to remove oxide film, while the second pulse is used for welding with a larger current.

### 3.5. Brazing

Xia et al. (2017) [40] studied vacuum brazing for an overlap joint of 50Mo-50Re alloy with a thickness of 0.06 mm. BM is prepared by utilizing powder metallurgy. The brazing filler metal is Ni-Cr-Si-B (Ni-19Cr-7.3Si-1.5B wt%) with a melting temperature range of 1081 to 1136 °C. After heat preservation for 20 min at brazing temperature of 1200 °C, the well-formed brazing seam was obtained, without defects, such as microcracks and pores. However, CrB and NiSi2 brittle intermetallic compounds are formed at the center of the brazing seam. Song et al. (2015) [41] studied vacuum brazing for overlap joint of TZM alloy (Ti 0.50 wt%, Zr 0.08 wt%, and C 0.04 wt%) with thickness of 3 mm. The brazing filler metal is Ti-28Ni (wt%) eutectic filler metal with melting temperature in the range of 940 to 980 °C. The range of brazing temperature is 1000–1160 °C and vacuum degree is

~1.33 MPa. The shear strength of the brazing joint preserved for 600 s at 1080 °C reaches ~107 MPa. The shear fracture shows the morphology of quasi-cleavage transgranular fracture.

*3.6. Friction Welding*

Fu et al. [42] studied friction welding of Mo alloy and die steel. The results demonstrate that thermal coupling during the friction welding process is conducive to grain refinement near seam and closure of pores in TZM powder alloy. According to strict welding specifications, a good joint without defects can be obtained through friction welding. Yazdanian et al. [43] researched friction stir welding of pure Mo plate (99.5 wt%) with thickness of 1.5 mm by utilizing a stir-welding head of Iridium (Ir)-Re alloy. The strength of butt welded joint prepared at rotation speed of 1000 rpm and welding speed of 100 mm/min reaches 86% that of BM and the joint is fractured in HAZ in a tensile test. Reheis et al. (2014) [44] investigated continuous drive friction welding of a TZM alloy tube with an outer diameter of 55 mm and wall thickness of 7.5 mm. A well-formed joint is obtained under the optimized process parameters, and its tensile strength at room-temperature is equivalent to that of BM, whereas its elongation is ~50% lower than that of BM. Ambroziak et al. (2011) [45] studied continuous drive friction welding of a refractory metal bar with the diameter of 30 mm under different combinations, such as Mo-Mo, TZM-TZM, TZM-V, TZM-Ta, Mo-Nb, and TZM-NB. In the whole welding process, the sample was immersed in IME82 oil (Figure 17) to prevent the workpiece from being polluted by environmental gas at high temperature. The results present that the well-formed welded joint with fine grains can be obtained for each combination under reasonable technological conditions.

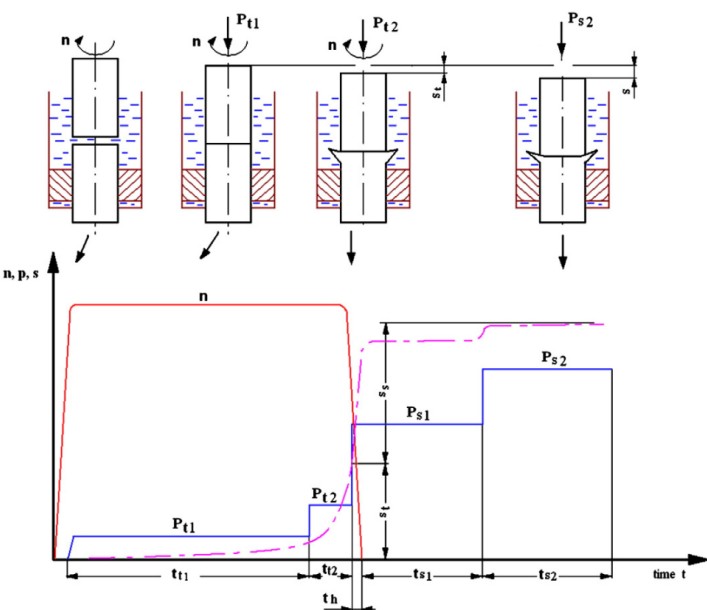

**Figure 17.** Scheme of friction welding in IME82 oil [45].

Recently, Stütz et al. (2016) [46] realized continuous drive friction welding of a pure Mo tube with wall thickness of 10 mm and outer diameter of 130 mm (Figure 18). In the study of Stutz et al. (2018) [47,48], continuous dynamic recrystallization and the competing dynamic recovery were observed as key mechanisms; intensive subgrain formation and the onset of recrystallization played the major role on the microstructure modification due to rotary friction welding. Grain refinement is observed in the weld interface for the TZM, whereas coarse grains are observed in the same zone for pure Mo but comparable crystallographic texture is observed for both materials, the result is shown in Figure 19.

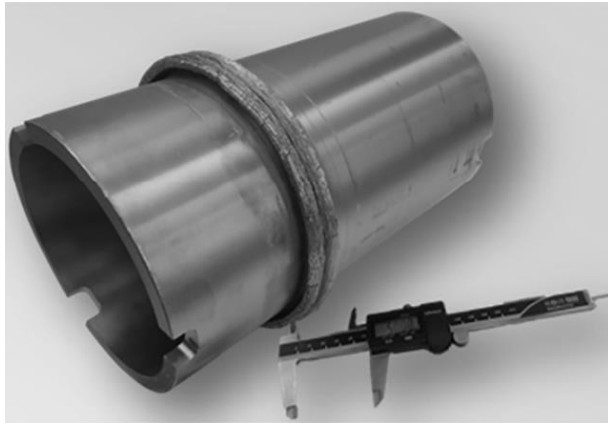

**Figure 18.** Medium-size friction welded Mo-tube (OD: 150 mm, ID: 130mm) [46].

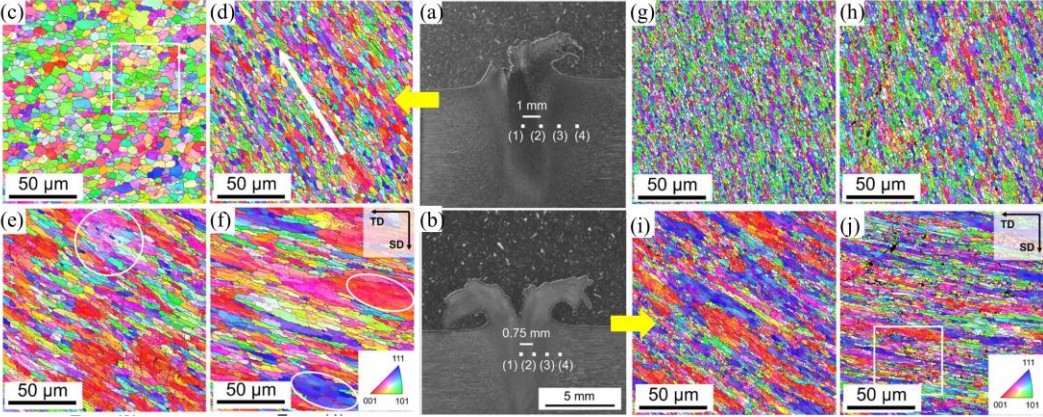

**Figure 19.** Detail of the microstructure friction welded samples: Overview of (**a**) Mo; overview of (**b**) TZM; (**c**) zone (1) of panel (**a**); (**d**) zone (2) of panel (a); (**e**) zone (3) of panel (**a**); (**f**) zone (4) of panel (**a**); (**g**) zone (1) of panel (**b**); (**h**) zone (2) of panel (**b**); (**i**) zone (3) of panel (**b**); (**j**) zone (4) of panel (**b**). [48].

## 4. Summary and Prospects

### 4.1. Advantages and Disadvantages of Various Methods for Welding Mo and Mo Alloys

Mo and Mo alloys are commonly used as high-temperature structural materials under harsh working conditions. Due to low welding temperature and uniform heating of weldment, brazing incurs small deformation and is easy to ensure the accurate size of weldment. However, strength and heat resistance of brazing seam are lower than those of BM and its performance at high temperature is generally inferior to that of joints obtained by fusion welding. High electric conductivity and yield strength at high temperature of Mo worsen its weldability in ERW. In friction welding, the instrument is worn seriously and an irreparable keyhole is formed in the weld seam when lifting the stir-welding head out from the workpiece after welding. Moreover, corrosion resistance of the weld seam reduces and it is difficult to clamp the components of the thin-walled tube. The high thermal conductivity, the significant tendency of grain coarsening, and grain-boundary embrittlement of Mo alloys determining that welding using a heat source with high power density has great advantages for Mo and Mo alloys. EBW with high energy density can be used for welding refractory alloys and difficult-to-weld materials, and shows fast welding speed, small HAZ, and small welding stress and deformation. The vacuum environment for EBW can not only prevent the molten metal from being polluted by gases, such as oxygen and nitrogen, but also facilitate the degassing and purification of metal in weld seam. Therefore, EBW is widely used in welding of Mo and Mo alloys. However, EBW

has shortcomings, such as complex process, low efficiency, limited size and shape of weldment by the vacuum chamber, and susceptibility to the interference of stray electromagnetic field and X-ray radiation produced during welding. In recent years, fiber laser and Disk laser technologies with excellent beam quality have developed rapidly, providing opportunities and necessary conditions for the breakthrough and development of welding technologies for Mo and Mo alloys. Laser welding not only has many advantages afforded by heat sources with high energy beams, such as high-power density and small heat input, but also can be carried out in an open environment. In the meanwhile, it is more difficult to protect the high temperature zone and control pore defects in weld seam during laser welding of Mo and Mo alloys compared with EBW. In addition, note that the last welding spot of the nuclear fuel rod needs to be welded under hyperbaric environment to encapsulate the hyperbaric inert gas, so the mechanism and technology of laser welding molybdenum alloy under hyperbaric environment is a blank field to be studied urgently.

## 4.2. Demands and Prospects

In recent years, ATF fuel has gained attention in technical research and development in the international community of nuclear energy. Nuclear energy agencies in these countries, such as the United States, France, South Korea, and Japan, take ATF as their next key development direction. In 2016, Professor Stephen Zinkel, an academician of the National Academy of Engineering in the United States, pointed out that ATF fuel is profoundly influencing the development direction of science and technology of nuclear energy and will change "game rules" for nuclear safety and nuclear power industry in the world. High-performance Mo alloy is a main alternative material for the next generation of ATF cladding materials. Although it has excellent ductility, its weldability remains to be solved, so development of reliable welding technology for Mo alloy has become a pressing need. Studying problems, such as embrittlement and pore defects in welding of Mo and Mo alloys and exploring new methods and mechanisms for controlling welding quality, not only has important theoretical significance, but also shows important engineering application significance.

**Author Contributions:** Q.Z. and M.X. wrote the major part of the review; X.S., G.A., J.S., N.W., S.X., C.B., and J.Z. participated on the concept of review and made the corrections/suggestions for improvement. All authors have read and agreed to the published version of the manuscript.

**Funding:** This work was supported by the National Natural Science Foundation of China (Grant No. 51775416).

**Conflicts of Interest:** The authors declare no conflict of interest.

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
