# Peer review of "Research Status and Progress of Welding Technologies for Molybdenum and Molybdenum Alloys"

_metals, doi:10.3390/met10020279_

Round 1

Reviewer 1 Report

The review paper is timely and may be of interest to the readership however several important flaws were observed during my review. The abstract of the article appears to be too short and does not capture the key/very important developments that have taken place in this area.

The introduction section also appears incomplete and lacking in numerous citations. In fact, no citations were added between lines 1 and 73 to substantiate the claims made. The structure of the paper require extensive improvements possible with the additions of images and schematics as well as microstructure and property details for Mo samples joined/welded.

The references section should also be extended to properly cover the field.

Author Response

Detailed responses to reviewers 1

Title: Research status and progress of welding technologies for molybdenum and molybdenum alloys

Response to Reviewer’s comments (highlighted with green in manuscript)

Thank you for your time and professional suggestions.

(1)

Comments:

The abstract of the article appears to be too short and does not capture the key/very important developments that have taken place in this area.

Response:

Agree.

It has been modified.

Please see: Lines 14-27.

(2)

Comments:

The introduction section also appears incomplete and lacking in numerous citations. In fact, no citations were added between lines 1 and 73 to substantiate the claims made.

Response:

Agree.

It has been modified.

Please see: Lines 33、37、46、48、50、57、64、74、90.

(3)

Comments:

The structure of the paper require extensive improvements possible with the additions of images and schematics as well as microstructure and property details for Mo samples joined/welded. The references section should also be extended to properly cover the field.

Response:

Agree.

It has been modified.

Please see: Lines 76、78、91、141、158、194、211、215、225、227、236、243、262、267、293、307、346、355、357.

Reviewer 2 Report

The review article needs more work before being accepted for publication.

My comments are as follows:

Extensive re-writing of the manuscript is is required to improve the language and style. No figures were included from any of the referenced manuscripts. Including figures will improves the readability of the manuscript as a whole. The introduction is very short, and is not elaborate enough. Please rewrite the introduction to clearly explain what is lacking, and why such a review of welding technologies of Mo and its alloys is important. Section 2.1: The authors discuss about porosity and powder metallurgy route for processing of Mo alloys, in line 65 the authors discuss about welding. The sentence is not coherent with the rest of the section. Line 83-85: It is not clear by what the authors mean by the bottom of the weld seam. Were different cross-weld tensile samples taken from the bottom to the top and then tensile tested? Please clarify. Line 86-87: More critical assessment is needed. Line 101-104 is a repetition of line 76-78. What is the new finding by Stutz et al. compared to Pan et al? Line 112: How does the morphology change. Please explain. Line 117: It would be beneficial for the reader, if the optimized parameters used were tabulated. Line 120: Repetition to Section 2.1. Please explain what is the new finding by the different authors of each study. Line 122: How addition of C improves plasticity? In line 91, the authors mention that carburization improves strength. Please explain how carburization/addition of C improves both strength and plasticity. Line 145: It is not clear what the authors mean by dark compounds. Are they precipitates/intermetallics? Line 159: The sentence is not clear. What do the authors mean by 1.5 times of weld seam by using two welding methods. Please clarify. Line 169: What are NS Mo joints? Line 187: Not clear. Do the authors mean increased by 280 MPa? If so what is the base value or with respect to which value is there a 280 MPa increase.

Author Response

Detailed responses to reviewers 2

Title: Research status and progress of welding technologies for molybdenum and molybdenum alloys

Response to Reviewer’s comments (highlighted with yellow in manuscript)

Thank you for your time and professional suggestions.

(1)

Comments:

Extensive re-writing of the manuscript is is required to improve the language and style.

Response:

Agree.

It has been modified.

Please see: Lines 49、54、58、62、67、69、70、82、83、98、100、101、103、104、105-107、109、118-120、121-122、123、129、137、153、161、164-165、169、185、190、191、192、198、199、202、210、222、241、249、253、279、283、297、303、316、318、330、334、335、337、353、362、364、371、379、384、387-390、394、395、400.

(2)

Comments:

No figures were included from any of the referenced manuscripts. Including figures will improves the readability of the manuscript as a whole.

Response:

Agree.

It has been modified.

Please see: Lines 76、78、91、141、158、194、211、215、225、227、236、243、262、267、293、307、346、355、357.

(3)

Comments:

The introduction is very short, and is not elaborate enough. Please rewrite the introduction to clearly explain what is lacking, and why such a review of welding technologies of Mo and its alloys is important.

Response:

Agree.

It has been modified.

Please see: Lines 36-46.

(4)

Comments:

Section 2.1: The authors discuss about porosity and powder metallurgy route for processing of Mo alloys, in line 65 the authors discuss about welding. The sentence is not coherent with the rest of the section.

Response:

Agree.

This paper reviews the welding technology of molybdenum alloy, the section 2.1 of manuscript discussions about poverty and crowd metrology route for processing of Mo alloys is to explain that the pore defects of Mo alloys have a bad impact on their welding quality, in response to the "the sentence is not coherent with the rest of the section" mentioned by the reviewer.

Please see: Lines 74-75.

(5)

Comments:

Line 83-85: It is not clear by what the authors mean by the bottom of the weld seam. Were different cross-weld tensile samples taken from the bottom to the top and then tensile tested? Please clarify.

Response:

Agree.

There is a problem in our statement here. We will change it to "was found in the weld seam".

Please see: Lines 109.

(6)

Comments:

Line 86-87: More critical assessment is needed.

Response:

Agree.

Here it is modified to "As can be seen from the above research results, the results illustrate that grains in weld seam of Mo by EBW grow rapidly.”

Please see: Lines 112-113.

(7)

Comments:

Line 101-104 is a repetition of line 76-78. What is the new finding by Stutz et al. compared to Pan et al?

Response:

Agree.

The results of Stutz et al. show that the porosity defects are serious when the heat input is large. The small heat input can not only inhibit the porosity but also significantly reduce the grain size of FZ. Pan et al. shows that the faster the welding speed is, the smaller the grain size is, and the less the impurities in the gap are. By increasing the welding speed and reducing the welding heat input, the ductility of Mucun welded joint can be significantly improved.

Both of them have shown that the heat input during the welding process should be reduced.

Please see: Lines 105-107.

(8)

Comments:

Line 112: How does the morphology change. Please explain.

Response:

Agree.

Compared with the columnar crystal during welding without beam deflection, the microstructure of the weld zone transforms to an equiaxed crystal when welding with a beam offset to Kovar. The morphology of the reaction layer alters due to the beam deflection, which escalates its toughness.

Please see: Lines 146-149.

(9)

Comments:

Line 117: It would be beneficial for the reader, if the optimized parameters used were tabulated.

Response:

Agree.

Better welding process parameters: welding speed 4 mm/s, argon flow rate 10 L/min, welding current should be controlled at about 210 A.

Please see: Lines 155-156.

(10)

Comments:

Line 120: Repetition to Section 2.1. Please explain what is the new finding by the different authors of each study.

Response:

Agree.

Section 2.1 is just a question, but the relevant studies here try to solve the problem of TIG welding of molybdenum alloy through research.

(11)

Comments:

Line 122: How addition of C improves plasticity? In line 91, the authors mention that carburization improves strength. Please explain how carburization/addition of C improves both strength and plasticity.

Response:

Agree.

First, solid carburizing (SC) can achieve the goal of addition C to joint and C distributes in the FZ of Mo in the form of C atom and Mo2C. Second, grain boundary strength and ductility in the grain are improved due to SC. The coordination between grain and grain boundary is improved during deformation. In the end, Mo2C can prevent the crack propagation during the deformation.

[Liang-Liang Zhang, Lin-Jie Zhang, Jian Long, Xu Sun, Jian-Xun Zhang, Suck-Joo Na. Enhanced mechanical performance of fusion zone in laser beam welding joint of molybdenum alloy due to solid carburizing [J]. Materials and Design, 2019, 181: 107957.]

(12)

Comments:

Line 145: It is not clear what the authors mean by dark compounds. Are they precipitates/intermetallics?

Response:

Agree.

For the sake of accuracy, we add the following explanation here, “the composition analysis showed that the content of C and O in these compounds were 30% and 15%, respectively.”

Please see: Lines 187-189.

(13)

Comments:

Line 159: The sentence is not clear. What do the authors mean by 1.5 times of weld seam by using two welding methods. Please clarify.

Response:

Agree.

For better understanding, change here to “The weld width of EBW and hybrid welding method is about 1.4mm and 2.6mm respectively, but in both cases, the width of HAZ is about 1.5 times of the weld width.”

Please see: Lines 205-207.

(14)

Comments:

Line 169: What are NS Mo joints?

Response:

Agree.

NS Mo is the abbreviation of novel molybdenum alloy with Nano-sized rare earth oxide particles and Superfine crystal microstructure, we give comments in the text.

Please see: Lines 231-232.

(15)

Comments:

Line 187: Not clear. Do the authors mean increased by 280 MPa? If so what is the base value or with respect to which value is there a 280 MPa increase.

Response:

Agree.

The tensile strength without Ni element is 112mpa, and it can reach 280mpa after adding. In order to express more clearly, it is modified as "The results demonstrated that tensile strength of the joints could be increased to about 280 MPa by presetting a nickel (Ni) foil at Mo/304L interface and shifting laser beam to 304L, and the tensile strength of the sample without Ni foil is only 112mpa."

Please see: Lines 272-274.

Round 2

Reviewer 1 Report

The authors have now included nineteen new figures, however none of the figure are discussed in the accompanying paragraphs/text. These images should support the discussion.  The caption for Figure 2 should be divided into (a) and (b), however (a) is missing.  Similarly, the labels for all the Figures from 1-19 (a).....(f) should be explained in the figure caption. Figure 19-please use letters to label each image instead of numbers. 

Author Response

Detailed responses to reviewer 1

Title: Research status and progress of welding technologies for molybdenum and molybdenum alloys

Response to Reviewer’s comments (highlighted with yellow in manuscript)

Thank you for your time and professional suggestions.

(1)

Comments:

The authors have now included nineteen new figures, however none of the figure are discussed in the accompanying paragraphs/text. These images should support the discussion.

Response:

Agree.

It has been modified.

Please see: Lines 70、72、88、141-142、158、191-192、212-213、228、237-238、247-248、259-260、270、283-284、315-316、352、359、364-365.

(2)

Comments:

The caption for Figure 2 should be divided into (a) and (b), however (a) is missing.  Similarly, the labels for all the Figures from 1-19 (a).....(f) should be explained in the figure caption. Figure 19-please use letters to label each image instead of numbers.

Response:

Agree.

It has been modified.

Please see: Lines 78、81、94、146、163、199、217、221、233、243、251、272、277、304、369.

Round 3

Reviewer 1 Report

I believe the article can now be accepted for publication.

Author Response

Detailed responses to reviewer 1

Title: Research status and progress of welding technologies for molybdenum and molybdenum alloys

Response to Reviewer’s comments (highlighted with yellow in manuscript)

Thank you for your time and professional suggestions.

(1)

Comments:

Abstract

Owing to its potential application prospect in novel accident tolerant fuel, molybdenum (Mo) alloys and their welding technologies has gained great importance in recent years.

Comment: According this sentence, Mo is an ingredient of accident tolerant fuel.

Response:

Agree.

It has been modified.

Please see: Lines 14-15.

(2)

Comments:

“...electron-beam welding (EBW), tungsten-arc inert gas (TIG) welding, laser welding, electric resistance welding (ERW), brazing and friction welding in welding Mo and Mo alloys,...”

Comment: Since you provide acronyms for other methods, provide also the acronym for laser welding (LW). You listed different methods of joining, not just welding. Therefore, the sentence must be modified as follows : ...brazing and friction welding in joining Mo and Mo alloys,...

Response:

Agree.

It has been modified.

Please see: Lines 24-25.

(3)

Comments:

Introduction

“... favorable heat and electrical conductivity...”

Comment: What do you mean favorable heat and electrical conductivity? It is high or low?

Response:

Agree.

It has been modified.

Please see: Lines 32.

(4)

Comments:

“small coefficient of linear expansion”

Comment: The proper is low coefficient.

Response:

Agree.

It has been modified.

Please see: Lines 32-33.

(5)

Comments:

“Oxygen is the most important impurity element in the gap which affects..”

Comment: What do you mean in the gap?

Response:

Agree.

“in the gap” refers to the “grain boundary”, in order to better understand, we modify this expression. It has been modified.

Please see: Lines 39.

(6)

Comments:

“Mo has advantages, such as small neutron absorption cross section, excellent high-temperature strength, good thermal conductivity, small coefficient of linear expansion and good wear and corrosion resistance.”

Comment: The above sentence is almost repetition of the first sentence in the Introduction. Therefore, it is recommended to remove it.

Response:

Agree.

It has been remove.

Please see: Lines 53.

(7)

Comments:

“Therefore, Mo alloy is listed as one of the main candidate materials for ATF cladding by the global nuclear industry [8]. In this context, the welding technologies for Mo...”

Comment: If you refer to one Mo alloy, please indicate which alloy you mean. Do you refer to cladding or welding? These are two different groups of technologies characterized by different challenges.

Response:

Agree.

The molybdenum alloy here is not of a specific brand. The nuclear industry has mentioned the introduction of molybdenum alloy into the next generation of alternative materials for ATF cladding tubes, but has not emphasized which brand, and in this refers to cladding material.

(8)

Comments:

Analysis on weldability of Mo and Mo alloy

“Ductility of Mo and Mo alloy varies...”

Comment: What kind of Mo alloy do you mean? Verify it throughout the text. Response:

Agree.

This is to explain that most of the molybdenum and molybdenum alloy ductility changes with the change of temperature, not a specific brand. In order to better understand, we modify this expression.

Please see: Lines 59.
